# Dietary Supplementation of a Commercial Prebiotic, Probiotic and Their Combination Affected Growth Performance and Transient Intestinal Microbiota of Red Drum (*Sciaenops ocellatus* L.)

**DOI:** 10.3390/ani12192629

**Published:** 2022-09-30

**Authors:** Fernando Y. Yamamoto, Matthew Ellis, Paul R. Bowles, Blaine A. Suehs, Pedro L. P. F. Carvalho, Caitlin E. Older, Michael E. Hume, Delbert M. Gatlin

**Affiliations:** 1Department of Wildlife and Fisheries Sciences, Texas A&M University, College Station, TX 77845, USA; 2Thad Cochran National Warmwater Aquaculture Center (NWAC), MAFES, Mississippi State University, Stoneville, MS 38776, USA; 3Department of Ecology and Conservation Biology, Texas A&M University, College Station, TX 77845, USA; 4USDA, ARS, Warmwater Aquaculture Research Unit, Stoneville, MS 38776, USA; 5USDA, ARS, SPARC, Food and Safety Research Unit, College Station, TX 77845, USA

**Keywords:** feed additives, next-generation sequencing, synbiotic, transient intestinal microbiome

## Abstract

**Simple Summary:**

Feeds for farmed fish containing high levels of plant ingredients and high stocking densities can impact the marine carnivorous fish growth and impair their intestinal health status, which ultimately leads to an increased infection susceptibility. To mitigate these negative impacts, feed additives such as live bacteria (probiotics), and complex fibers to induce the growth of beneficial bacteria in the intestine (prebiotics), have been successfully supplemented to the diets of different fish species, including the red drum. This is the first report to date exploring potential synergisms between prebiotic and probiotic for this important marine fish species. Our results show that these additives in low fishmeal-based diets can improve growth performance and modify the protein composition of red drum juveniles. Their transient intestinal bacterial community was also modulated, by reducing the abundance of possible pathogenic bacteria responsible for considerable losses in intensive aquaculture systems. These findings are important steps towards more sustainable aquaculture practices and environmentally oriented aquafeed formulations.

**Abstract:**

In the present study, the potential synergism between beneficial lactic acid bacteria (*Pediococcus acidilactici*) contained in a probiotic and a mixture of fermentable complex carbohydrates and autolyzed brewer’s yeast (or prebiotic) were explored in red drum. Four experimental diets were formulated from practical ingredients, and the basal diet was supplemented with either probiotic, prebiotic, or both supplements. Red drum juveniles (~5.5 g) were offered the four experimental diets for 56 days, and at the end of the feeding trial fish fed diets supplemented with probiotic had significantly better weight gain than those fed the non-supplemented diets, and higher protein content in their whole-body composition. Transient intestinal microbiome alpha and beta diversity were significantly affected by the dietary treatments. Interestingly, a higher relative abundance of the lactic acid genus *Pediococcus* was observed for fish fed diets supplemented with the prebiotic. A higher relative abundance was also observed for the predicted functions of the microbial metagenome, and many of these pathways involved the biosynthesis of essential amino acids, vitamins, and nucleotides. Even though no potential synergistic effect was observed, the individual inclusion of these prebiotic and probiotic supplements positively affected the intestinal health and growth performance of red drum, respectively.

## 1. Introduction

The advent of feed additives to modulate immune responses and enhance intestinal health has been an established strategy in animal husbandry to diminish losses caused by disease outbreaks, and this is especially true for fed-aquaculture [1]. The current aquafeed formulation trends, where minimum forage fish ingredients are being included in modern carnivorous fish diets, combined with the increased stocking densities to improve water usage efficiency, can lead to farmed carnivorous fish being increasingly subjected to chronic stress conditions [2]. Alike any other animal husbandry, the interrelationship between fish stress and disease resistance plays a pivotal role in their ability to combat infections. To mitigate these issues and invigorate the intestinal health of farmed fish, the use of feed additives such as prebiotics and probiotics or their combination (synbiotic) can be a sustainable prophylactic approach to inhibit pathogen proliferation and modulate immune responses by manipulating the commensal intestinal microbiome, and thereby minimize losses due to epizootic outbreaks [3].

Prebiotics are functional polysaccharides that are not readily digestible by the host, but they can serve as nourishment by the natural commensal intestinal microbiota [4]. Grobiotic^®^-A (International Ingredient Corporation, St. Louis, MO, USA) is a commercial prebiotic manufactured from partially autolyzed brewers’ yeast in combination with dairy components and dried fermentation products, with several studies reporting positive effects of its inclusion on production performance and disease resistance for a variety of fish species [5]. Probiotics are defined as living microorganisms that can benefit the host by improving the intestinal microbial balance and by inhibiting the growth of pathogenic microorganisms, increasing feed nutrient utilization, and stimulating the immune response of the host [1,6]. Bactocell^®^ (Lallemand, Montreal, QC, Canada) is a monogastric feed additive, mainly composed of living lactic acid bacteria (*Pediococcus acidilactici*), that can ferment non-digestible carbohydrates into lactic acid. It also has been shown to improve the production performance and immunological responses of several fish species [6]. The concept of supplementing prebiotics and probiotics together as synbiotics foresee the interaction between these two supplements, i.e., the fermentation of the prebiotic compounds by the live bacteria from the probiotic, yielding a synergistic beneficial effect on the host [3].

Red drum (*Sciaenops ocellatus*) is an euryhaline sciaenid species with desirable characteristics for aquaculture [7]. This species can easily transition from consuming live food to inert diets during their juvenile stages, and their broodstock spawning technology is well-established, and then allowing the production of fingerlings year-round [8]. Even though the red drum has a stenophagous feeding behavior in the wild, in captivity, it can readily consume diets containing low levels of fish-derived ingredients without any apparent detrimental effects [9,10,11]; nevertheless, it is still paramount to investigate strategies to improve red drum nutrient uptake, modulate their immune responses and balance their intestinal microbiota when fed diets formulated with low levels of marine derived ingredients. The objective of this study was to evaluate the dietary supplementation of Bactocell^®^ and Grobiotic^®^-A individually or their combination on juvenile red drum.

## 2. Materials and Methods

### 2.1. Fish, Experimental Diets, and Culturing Conditions

Experimental diets were formulated to contain 420 g of crude protein and 120 g of crude lipid per kg on a dry-matter basis, and a calculated digestible energy of 14.2 MJ per kg of diet (Table 1). The commercial prebiotic (Grobiotic^®^-A) was included in the formulation at 20 g kg^-1^ the expense of cellulose and glycine in the basal diet to ensure the experimental diets were isonitrogenous. The commercial probiotic (Bactocell^®^) was included at 1 g kg^−1^ (or 1 × 10^9^ CFU kg^−1^), as recommended by the manufacturer, at the expense of cellulose only. The experimental diet containing both prebiotic and probiotic (Pre + Pro) had the same inclusion levels as the individual treatments at the expense of cellulose and glycine. The experimental diets were mixed and cold pelleted using established methods from our laboratory as previously described [12], and their nutrient composition was analyzed following the AOAC procedures [13].

Red drum fingerlings were obtained from the Texas Parks and Wildlife Department hatchery at Lake Jackson, TX, and quarantined in a 250-L fiberglass tank operating as a recirculating system. The fingerlings were trained to consume inert diets using a commercial feed formulated for carnivorous marine fish (Otohime C1, Marubeni Nisshin Feed Co., Tokyo, Japan). When the fish reached the appropriate size and weight, they were moved to 38-L aquaria, also operating as a recirculating system, and fed the basal diet for a week conditioning period until the commencement of the experiment. All procedures performed in the present study were in compliance with the Institutional Animal Care and Use at Texas A&M University, and this investigation was approved under the protocol IACUC-2019-0448.

The recirculating system where the feeding trial was conducted comprised a common settling chamber for sedimentation of solids, along with biological, mechanical, and ultraviolet systems. Aeration was provided to the tanks with individual air stones to each aquarium and to the biofilter through a regenerative air blower. The water temperature was conditioned by ambient air, and water alkalinity, hardness, and salinity were kept at minimum levels by adding synthetic marine salt (Red Sea Salt, Red Sea, Houston, TX, USA). A photoperiod was set for 12 h light and 12 h dark using fluorescent lights controlled by timers. A total of 192 fish with an initial weight of 5.59 ± 0.02 g (average ± standard deviation; SD) were equally distributed in 16 aquaria (12 fish/aquarium), and the four experimental diets were randomly assigned to four replicate tanks (*n* = 4) in a fashion that the variation proportioned by the tank disposition could be accounted for and statistically blocked. A subsample of the fish was euthanized with an overdose of tricaine methanesulfonate (MS-222 at 300 mg L^−1^, Western Chemical, Ferndale, WA, USA) [14] at the initiation of the trial to estimate the initial nutrient composition of the whole-body. Fish were group-weighed weekly, and the feeding rate was fixed throughout the week and adjusted as the fish grew according to a percentage of the tank biomass (from 6% to 2.75%), which would be close to apparent satiation. Fish were fed twice daily with the prepared rations for 8 weeks.

Water quality parameters were monitored three times a week to ensure that culture conditions were appropriate for red drum culture [15]. Dissolved oxygen, pH, and salinity were measured as previously described [12], and total alkalinity and hardness were measured with commercial kits (Hach Company) using sodium hydroxide-phenolphthalein, and EDTA titration, respectively. Ammonia-nitrogen and nitrite-nitrogen were measured photometrically with test reagents and the DR2000 spectrophotometer as specified by the manufacturer (Hach Company). Water quality parameters measured throughout the feeding trial were reported as follows (average ± SD): Temperature = 26.7 ± 0.7 °C; dissolved oxygen = 6.95 ± 0.97 mg L^−1^; pH = 8.24 ± 0.27; salinity = 2.55 ± 0.91 mg L^−1^; total alkalinity = 143.5 ± 27.6 mg CaCO_3_ L^−1^; total hardness = 368.6 ± 148.5 mg CaCO_3_ L^−1^; total ammonia-nitrogen = 0.11 ± 0.12 mg L^−1^; and total nitrite-nitrogen = 0.027 ± 0.024 mg L^−1^.

### 2.2. Sampling Procedures

After the 8 weeks of feeding, all fish from each aquarium were counted and group weighed to compute weight gain. Each tank had three fish randomly selected and euthanized with MS-222 using the methods previously described for whole-body proximate composition. An additional set of three fish were anesthetized (100 mg L^−1^ of MS-222), and blood samples were collected with heparinized tuberculin syringes. The same fish were then euthanized with MS-222, individually weighed, and the contents on the intraperitoneal cavity were carefully dissected to sample the liver and the intraperitoneal fat (IPF) to compute the viscerosomatic indices (HSI or IPF). Calculations were as follows:$$\mathrm{Percentage}\;\mathrm{of}\;\mathrm{weight}\;\mathrm{gain}\;\left(\%\;\mathrm{of}\;\mathrm{initial}\right)=100\;\times \;\;\left[\left(\mathrm{Average}\;\mathrm{weight}\;\mathrm{at}\;\mathrm{the}\;8\mathrm{th}\;\mathrm{week}\;(g)\;-\;\mathrm{average}\;\mathrm{initial}\;\mathrm{weight}\;(g)\right)/\mathrm{average}\;\mathrm{initial}\;\mathrm{weight}\;(g)\right]$$
$$\mathrm{Feed}\;\mathrm{efficiency}\;\left(\mathrm{FE}\right)=\mathrm{weight}\;\mathrm{gain}\;(g)\;/\mathrm{dry}\;\mathrm{feed}\;\mathrm{intake}\;(g)$$
$$\mathrm{Protein}\;\mathrm{conversion}\;\mathrm{efficiency}\;(\mathrm{PCE})\;\left(\%\right)=\{[(\mathrm{Final}\;\mathrm{body}\;\mathrm{weight}\;\left(g\right)\;\times \;\mathrm{final}\;\mathrm{body}\;\mathrm{protein}\;\left(\%\right)\;-(\mathrm{initial}\;\mathrm{weight}\;\left(g\right)\;\times \;\mathrm{initial}\;\mathrm{body}\;\mathrm{protein}\;\left(\%\right))]\;÷\;\mathrm{protein}\;\mathrm{intake}\;(g)\}\;\times \;100$$
$$\mathrm{Viscerosomatic}\;\mathrm{indices}\;\left(\mathrm{HSI}\;\mathrm{or}\;\mathrm{IPF}\;\mathrm{ratio}\right)\;\left(\%\right)=[\mathrm{liver}\;\mathrm{or}\;\mathrm{IPF}\;\left(g\right)/\mathrm{body}\;\mathrm{weight}\;\left(g\right)]\;\times \;100$$
$$\mathrm{Survival}\;\left(\%\right)=100\;\times \;(\mathrm{number}\;\mathrm{of}\;\mathrm{surviving}\;\mathrm{fish}/\mathrm{initial}\;\mathrm{number}\;\mathrm{of}\;\mathrm{fish})$$

The remaining fish were fed the experimental diets for an additional 4 days. On the 60th day of feeding, fish were fed to apparent satiation staggered in 10 min intervals between tanks to ensure that the intestinal digesta would be collected at a similar sampling time. At 5 h postprandial, three fish were randomly selected and euthanized with MS-222 as previously described. The intestinal tract from each fish was aseptically dissected, and the whole digesta content was squeezed out with sterilized tweezers into DNase- and RNase-free screw cap microtubes (one per tank). Each tubes with digesta samples were immediately flash-frozen in liquid nitrogen, and stored at −80 °C until further processed.

### 2.3. DNA Extraction from the Intestinal Contents, PCR, and Denaturing Gradient Gel Electrophoresis (DGGE)

DNA extraction from the samples followed the procedures previously described by Yamamoto et al. [12]. Briefly, samples were incubated with a lysing buffer (20 mg mL^−1^ lysozyme, 20 mM Tris-HCl; pH 8; 2.0 mM EDTA; 1.2% Triton-X) for 30 min at 37 °C. DNA samples were further isolated following the protocol of a commercial kit using the silica-membrane nucleic acid purification method (QIAamp DNA Mini Kit, cat# 51306, Qiagen, Valencia, CA, USA). DNA concentration from each sample was estimated via UV absorbance, and diluted to 50 ng of DNA µL^−1^. Samples were aliquoted, and a subset of all samples was shipped overnight to the University of Minnesota Genomics Center (UMGC) for next-generation sequencing.

The subset of samples kept was subjected to PCR using primers targeting the V3 region of the bacterial 16S rRNA gene. The methods for the PCR cycles, verification of the PCR products, DGGE gel preparation, and conditions for the electrophoresis were as previously described by Yamamoto et al. [12]. The comparison of samples band patterns was performed using Dice percentage similarity coefficient (%SC) and dendrograms were constructed using unweighted pair group with arithmetic averages method using Gel Compare II 6.6 (Applied Maths Inc., Austin, TX, USA).

### 2.4. Next-Generation Sequencing (NGS)

At UMGC, the DNA samples (*n* = 16) were subjected to amplification by targeting the V4 region of the 16S rRNA gene. Amplicons were sequenced using an Illumina MiSeq as described by Gohl et al. [16], but with 30 cycles for amplification. Raw fastq files are available to access at the NCBI Sequence Read Archive under the BioProject ID PRJNA736988. Adapters and primers were removed from the sequencing data using cutadapt. Data were further processed using QIIME2 (v. 2020.6) [17], where denoising was performed with DADA2 [18], and taxonomic classification was performed using scikit-learner classifier [19] trained on the 99% operation taxonomic unit (OTU) data set from the SILVA database release 132 [20] trimmed to the 515F/806R sequencing region [21]. To analyze alpha- and beta-diversity, samples were rarefied to 3020 sequences per sample, leading to the exclusion of one probiotic (TK24) sample. The predicted functional output of the transient intestinal microbiome was assessed using PICRUSt2 [22].

### 2.5. Blood Samples and Plasma Immunological Assays

Blood samples from each fish were individually stored in 1.5-mL microcentrifuge tubes and immediately centrifuged at 10,000× *g* for 10 min. Plasma samples were recovered using sterile pipette tips and stored frozen at −20 °C until further processed. Plasma total protein was estimated photometrically using Coomassie Blue (cat# 500-0006, Bio-Rad Laboratories, Hercules, CA, USA) by equivalent absorbance using a bovine serum albumin solution (cat# A2153, Sigma-Aldrich, St. Louis, MO, USA) as the standard curve. Samples were read on a microplate at 595 absorbance. Plasma total immunoglobulins also were estimated using Coomassie blue and subtracting the concentration from the respective total protein values after precipitating diluted samples with 12% polyethylene glycol (PEG, cat# P6667, Sigma Aldrich) [23]. Plasma lysozyme was evaluated by incubating the individual samples with a suspension of *Micrococcus lysodeikticus* (cat#M3770, Sigma-Aldrich Co.) at 0.2 g L^−1^ of phosphate-buffered saline as previously described [12].

### 2.6. Statistical Analysis

Data from the production performance, whole-body proximate, and plasma immune responses were analyzed as a 2 × 2 factorial design mixed with a block design. The disposition of the aquaria was considered as a statistical block. Statistical analyses were performed using JMP Pro software (v 14.0, SAS Institute, Cary, NC, USA). The presence or absence of prebiotic and probiotic were considered the main factors. If significant differences (*p* < 0.05) were detected for the main factors, a Student’s *t*-test was performed. If significant interactions between the main factors were detected, data would be subjected to a Tukey-HSD test.

For the NGS results, the output data were subjected to alpha diversity metrics and data were analyzed using JMP Pro software as previously described [12]. If significant differences were detected for the alpha diversity data, the dietary treatments were ranked using the non-parametric Kruskal–Wallis’s test. Beta diversity was calculated with Jaccard, Bray–Curtis, and weighted and unweighted UniFrac metrics. Distance matrices were statistically analyzed using the analysis of similarities test (ANOSIM), available in the vegan package in R software [24], and images to visualize the clusters were created using Emperor [25]. The differential relative abundance of the intestinal bacteria and predicted metabolic pathways (PICRUSt2) was tested using the linear discriminant analysis effect size (LEfSe) algorithm [26] available at the Huttenhower Lab Galaxy (https://huttenhower.sph.harvard.edu/galaxy/, accessed on 16 February 2021). LEfSe results were considered significant when a linear discriminant analysis (LDA) score of log 10 > 3.0 and *p* < 0.05, and for the predicted functional pathways were determined at a LDA score of log 10 > 2.5 and *p* < 0.05.

## 3. Results

### 3.1. Production Performance, Condition Indices, Whole-Body Proximate Composition, and Plasma Immune Responses

Red drum grew rapidly over the 8-week period with all dietary treatments increasing their initial weight by over 600%. Fish showed significant (*p* < 0.05) increase in weight gain when fed diets supplemented with the probiotic; whereas those fed the prebiotic had slight but significantly reduced weight gain (Table 2). No statistical differences (*p* > 0.05) were observed for feed efficiency, HSI, IPF ratio, or survival. Significant differences also were not observed for red drum whole-body moisture, lipid or ash, or protein conversion efficiency; however, red drum fed the diets supplemented with probiotic had higher whole-body protein composition (Table 3). No statistical differences among fish fed the various diets were detected for plasma lysozyme activity or plasma total immunoglobulins. However, fish fed diets without supplementation of the prebiotic had higher plasma total protein concentration than fish fed the diets with prebiotic.

### 3.2. Transient Intestinal Microbiota

DGGE results indicated that the transient intestinal bacterial community of fish fed the basal diet and fish fed diets supplemented with prebiotic were likely identical; whereas fish fed diets supplemented with the probiotic and Pre + Pro were very similar (Figure 1). Diversity analyses from NGS data indicated that bacterial communities in red drum digesta were heavily influenced by the inclusion of the pre- and/or probiotic. The two alpha diversity metrics that consider evenness, Shannon diversity index and Pielou’s evenness, indicated significant differences in alpha diversity based on diet (*p* = 0.01 and *p* = 0.009, respectively). Fish fed the basal diet appeared to have greater intestinal microbial diversity than the prebiotic and the Pre + Pro diets (Figure 2). Additionally, analysis of beta diversity indicated community structure was influenced by diet when quantitative (Bray–Curtis R = 0.39, *p* = 0.001; and weighted UniFrac R = 0.50, *p* = 0.001) metrics were used (Figure 3). These findings were not supported by the two qualitative beta diversity metrics used (Jaccard and Unweighted UniFrac).

Fish fed the diets containing the prebiotic and the probiotic were found to be dominated by Firmicutes (average relative abundance + SD; prebiotic = 84.60 + 4.88%, probiotic = 53.20 + 9.35%, Pre + Pro = 72.31 + 13.12%) followed by much lower relative abundances of Proteobacteria (prebiotic = 6.41 + 4.18%, probiotic = 27.49 + 7.78%, Pre + Pro = 15.57 + 10.29%). Fish fed the basal diet had almost equal relative abundances of Firmicutes (34.6 + 15.6%) and Proteobacteria (30.14 + 11.60%) (Figure 4). Differential abundance testing with LEfSe confirmed that Proteobacteria was found in significantly higher relative abundance in fish fed the basal diet (LDA score = 4.316, *p* = 0.0157). Several other taxa were found to be differentially abundant based on diet, including higher relative abundances of *Pediococcus* spp. in the digesta of fish fed the prebiotic- containing diet (LDA score = 4.75, *p* = 0.0045) (Figure 5). and higher relative abundances of Streptococcaceae in the digesta of fish fed the control diet (LDA score = 3.84, *p* = 0.03).

Microbiota sequencing data also were subjected to the prediction of functional pathways, followed by testing for significantly differentially abundant pathways. A total of 84 pathways were found to have significantly different abundance among diets (LDA score (log 10) > 2.5 and *p* < 0.05), and selected pathways of nutritional significance were displayed in Figure 6. Fish fed the basal diet presented communities associated with pathways involving arginine and vitamin biosynthesis and degradation of histidine. Several pathways involving biosynthesis of nucleotides and lysine and lactic fermentation were enriched in the intestinal microbiota of red drum fed diets supplemented with prebiotic. For the fish fed the probiotic higher relative abundance of pathways for the biosynthesis of essential vitamins and amino acids (isoleucine, methionine, arginine, histidine) were observed. No significant pathways were observed for the Pre + Pro group. The remainder of the significant pathways can be found in Appendix A.

## 4. Discussion

This study not only represents an additional evaluation of *Pediococcus acidilactici* (Bactocell^®^) as a dietary supplement for cultured fish, but it also is the first reported study exploring possible synergisms between prebiotic and probiotic supplements in red drum. Interestingly, improved weight gain was only observed for red drum fed diets supplemented with Bactocell^®^. This finding is corroborated by studies that also reported a superior growth performance of other cultured fish fed the same probiotic supplement, such as the Brook charr (*Salvelinus fontinalis*) [27], zebrafish (*Danio rerio*) [28], and Asian sea bass (*Lates calcarifer*) [29]. Contrasting with the present study, an improved weight gain was promoted by the interaction between Bactocell^®^ and assorted prebiotic compounds in Nile tilapia (*Oreochromis niloticus*) [30], Korean rockfish (*Sebastes schlegeli*) [31], Asian sea bass [29], and European sea bass (*Dicentrarchus labrax*) [32]. This synbiotic effect was not observed for red drum in this study, and neither it was observed in the production performance of totoaba (*Totoaba macdonaldi*), another marine sciaenid species subjected to a similar experimental design, but testing Aquablend^®^ (BIO-CAT, Troy, VA, USA) as the *Bacillus* spp.-based commercial probiotic [33].

When evaluating the supplementation of Grobiotic^®^-A per se, or other prebiotic supplements in red drum diets, several studies presented improved growth performance, nutrient digestibility, intestinal morphology, and/or disease resistance [34,35,36,37]. These discrepancies with the present study may be attributed to the reduced fishmeal in the diets along with a higher inclusion of soybean products. In support of this hypothesis, no improvement in production performance was observed for red drum, and largemouth bass (*Micropterus salmoides*) fed diets formulated with high inclusion of soybean products (at 640 g kg^−1^ and 58 g kg^−1^, respectively) and supplemented with this prebiotic [38,39,40]. On the other hand, the decreased production performance could have been caused by the replacement of glycine to keep the experimental diets isonitrogenous. It was previously reported that glycine supplementation improved growth performance of red drum and largemouth bass fed diets containing high levels of plant protein ingredients [40,41]. The improved growth performance of red drum fed diets containing Bactocell^®^ may be attributed to the high inclusion of soybean meal in the formulated diets. One of the promising features of the bacterium *Pediococcus adilactici* is the ability to ferment nonstarch polysaccharides and degrade possible antinutritional factors, which could have promoted a healthier intestine and improved nutrient absorption. To support this assumption, a higher protein content was found in the whole-body tissues of red drum fed the probiotic-supplemented diets, and a higher trend in the protein conversion efficiency for these fish also was observed.

The manipulation of immune responses through dietary treatments is a prophylactic management strategy of interest for aquaculture to ultimately reduce the incidence of diseases and prevent economic losses [42]. The concept of providing beneficial probiotic bacteria along with complex carbohydrates to serve as their nourishment could produce a synbiotic effect and benefit the intestinal immune system [3]. In the present study, the immune responses in the circulating plasma of red drum were unaffected by the dietary treatments, and it agrees with the lack of stimulated immune responses in the totoaba [33] and tilapia [43]. It is unclear whether the immune responses were found in a basal state at the time of the sampling, where no stimuli were triggering the immune cells, or if the supplements could be eliciting other immunological parameters than those evaluated. For instance, the expression of the IgT and inflammatory cytokine genes in the rainbow trout (*Oncorhynchus mykiss*) intestine were significantly elevated when the fish were exposed to diets supplemented with *Pediococcus acidilactici* for 2 and 4 weeks [44]. The expression of immune-related genes in the posterior intestine of European sea bass also was heavily affected by the dietary intervention of mannan oligosaccharides and Bactocell^®^ [32]. In humans, ingestion of this bacterial species resulted in a higher secretion of mucosal IgA by regulating the expression of interleukin-6 (IL-6) and IL-10 in dendritic cells [45]. It is deemed necessary to perform additional immunological and molecular analyses to verify if the supplementation of these additives is not affecting the red drum immune responses in other tissues (e.g., intestine, head-kidney, and other mucosal sites).

The present study relied on DGGE analysis to preview if there was a substantial shift in the transient intestinal bacterial population, and the results observed were consistent with previous reports for red drum evaluating Grobiotic^®^-A [39,46] and Bactocell^®^ [47] individually. The characterization of red drum transient intestinal microbiota by sequencing the V4 region of the 16S rRNA gene using NGS, showed that the bacterial community mainly consisted of the phyla Firmicutes, Proteobacteria, Actinobacteria and Bacteroidota (in decreasing order of relative abundance). This finding has been consistently reported in the intestinal tract of this species [12,39,48]. In the present study, the supplementation of these additives influenced alpha-diversity, with the transient intestinal microbiota of fish fed the basal diet presenting a higher diversity using the Shanon diversity index and Pielou’s evenness metrics, both of which are weighted by evenness (i.e., how similarly abundant different taxa are). However, the two metrics based on richness used, Chao1 index and observed OTUs, did not indicate there were differences in the number of different bacterial taxa observed. These results are intriguing and counter-intuitive when one of the premises of supplementing dietary prebiotics, probiotics, or synbiotics is to promote intestinal health by increasing intestinal microbial diversity. The reduction in microbial population diversity is usually observed in diseased fish when compared to healthy individuals [49], and this decrease also suggests the effectiveness of the protective barrier provided by the commensal microbiota could have been compromised [50]. However, similar to the present study, a lower diversity was observed for totoaba treated with diets containing moderate inclusion levels of soybean meal and supplemented with the prebiotic Agavin [51]. The high relative abundance of *Pediococcus* spp., as observed in the relative abundance taxa plot, may be the main driver of the reduced diversity for the intestinal microbiota of prebiotic and Pre + Pro fed red drum. Nevertheless, the reduced evenness observed in fish fed diets containing pre- and/or probiotics may not necessarily have a negative impact on fish health; further research involving exposure of fish to stressful conditions or bacterial challenge would be useful to evaluate if these communities can prevent colonization by pathogens. The intestinal bacterial community of the dietary treatments presented distinct clustering on the PCoA plots, indicating that the microbial community structure was affected by the feed additives. These distinct clusters also were observed for the totoaba intestinal microbiome when the fish were fed similar dietary treatments [33].

Despite not presenting synergistic effects for the other evaluated parameters, the high relative abundance of *Pediococcus* spp. observed in the prebiotic group, and the *Pediococcus acidilactici* seeded synbiotic treatment makes it evident that this genus can benefit and nourish from the Grobiotic^®^-A substrate. However, the limitation of Illumina MiSeq to accurately differentiate bacteria species precludes the conclusion of what *Pediococcus* species are proliferating in the Pre + Pro group and if this species is commensal to the red drum intestine or if it is the one delivered in this commercial probiotic. Corroborating with the present findings, a higher relative abundance of *Pediococcus* spp., was found in Nile tilapia [52], Atlantic salmon (*Salmo salar*) [53], and rainbow trout [44], when these species were fed diets containing this commercial probiotic. However, it should be highlighted the relative abundance of this bacterial genus was found in greater proportions in the red drum intestine than the aforementioned studies. In agreement with the present finding, a higher relative abundance of *Bacillus* spp. was found in the intestinal microbiota of totoaba fed the same commercial prebiotic product [33].

A higher relative abundance of *Streptococcaceae* as observed in the presented data was also reported for Atlantic salmon fed the control diet when compared to the *Pediococcus acidilactici* supplemented group [53]. A similar reduction in *Streptococcus* sp. was also observed for rainbow trout during fry stages [54] and as juveniles [44] when delivering the same probiotic supplementation. Streptoccocosis is a disease that afflicts aquaculture practices worldwide and can be caused by several bacterial species [55]. For the present study, Illumina MiSeq identified 4 different species of *Streptococcus* in the transient digesta of red drum fed the control diets were *S. parauberis*, *S. iniae*, and *S. agalactiae*, which are known by being common bacterial pathogens afflicting farmed fish [55,56]. Reducing the incidence of these bacteria in the fish gut by using these supplements can be beneficial and of interest, especially when the main port of entry of *Streptococcus* spp. is likely through the intestinal tissues [56,57]. Nevertheless, full-length or shotgun sequencing are warranted methods to properly validate our observation.

The predicted pathways of five out of the eleven essential amino acids reported for fish [58] were affected by the supplementation of *Pediococcus acidilactici* in the diets of red drum. Even though the experimental diets were formulated to completely meet the amino acid requirements of red drum, it can be hypothesized that this phenomenon may be aiding the growth performance observed in this dietary group. To corroborate the present findings, an increased amino acid absorption was reported for humans after consuming plant protein with the probiotic *Lactobacillus paracasei* [59], and for pigs fed diets supplemented with *L. reuteri* 1 [60]. Predicted pathways for lactic acid biosynthesis through fermentation and nucleotides syntheses were observed for red drum fed diets supplemented with Grobiotic^®^-A. Nucleotides are important dietary components and conditionally essential nutrients in high-plant protein diets for fish [61] and are reported to modulate their immune system [62], and lactic acid is a desired fermentation metabolite produced by the commensal bacteria offered [63]. Nevertheless, these observations are predicted from the bacterial metagenome, and quantitative analyses are deemed necessary to verify if these supplements positively affect the red drum intestinal health and nutrient absorption. Future studies are warranted to validate if the *Pediococcus acidilactici* supplementation can ameliorate the amino acid absorption and retention when offering a low fishmeal-based diet to red drum.

## 5. Conclusions

In conclusion, supplementing prebiotic, probiotic, and synbiotic additives in low fishmeal-based diets can improve production performance and modulate the transient intestinal microbiota and whole-body protein composition of red drum. In addition, the supplementation of these additives potentially reduces in the red drum intestine the relative abundance of the *Streptococcaceae* family, which encompasses several pathogenic species. However, the limited analysis performed in the present study did not result in observation of an immunological improvement in the plasma of this fish species due to these additives. Additional research is encouraged to evaluate other aspects of the immune response of red drum when offered these dietary supplements and further validate the findings of the predicted bacterial metagenome.

## Figures and Tables

**Figure 1 animals-12-02629-f001:**
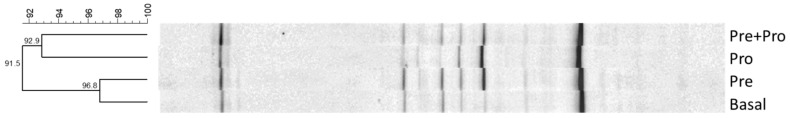
Denaturing gradient gel electrophoresis dendrogram of the red drum digesta microbiota. Percentage similarity coefficient (%SC, bar) ≤ 79% = not similar populations; %SC = 80–84% somewhat similar; %SC 85–89% similar; %SC = 90–94% very similar; and %SC ≥ 95% likely the same or identical.

**Figure 2 animals-12-02629-f002:**
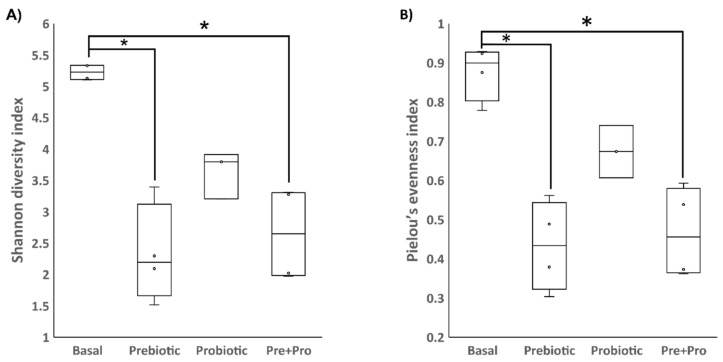
Comparing the alpha diversity using Shannon diversity (**A**) and Pielou’s evenness (**B**) indices to compare the transient intestinal microbiota of red drum fed the different dietary treatments. The bar and asterisks represent significant differences (*p* < 0.05) detected between the dietary treatments.

**Figure 3 animals-12-02629-f003:**
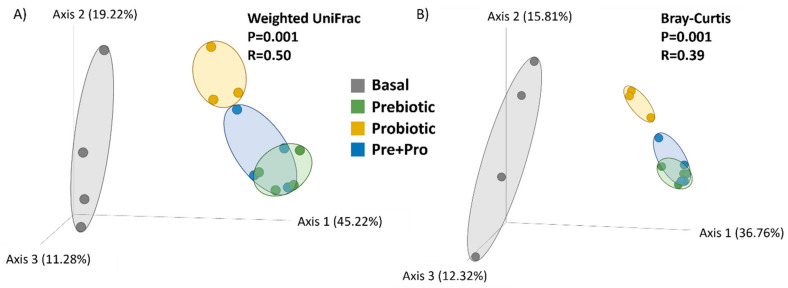
Principal Component Analysis (PCoA) plot based on the weighted UniFrac (**A**) and Bray–Curtis (**B**) as beta diversity metrics. Samples cluster by diet indicating the community structure is affected by diet, which was supported by the results of ANOSIM testing (R = 0.50, *p* = 0.001, respectively).

**Figure 4 animals-12-02629-f004:**
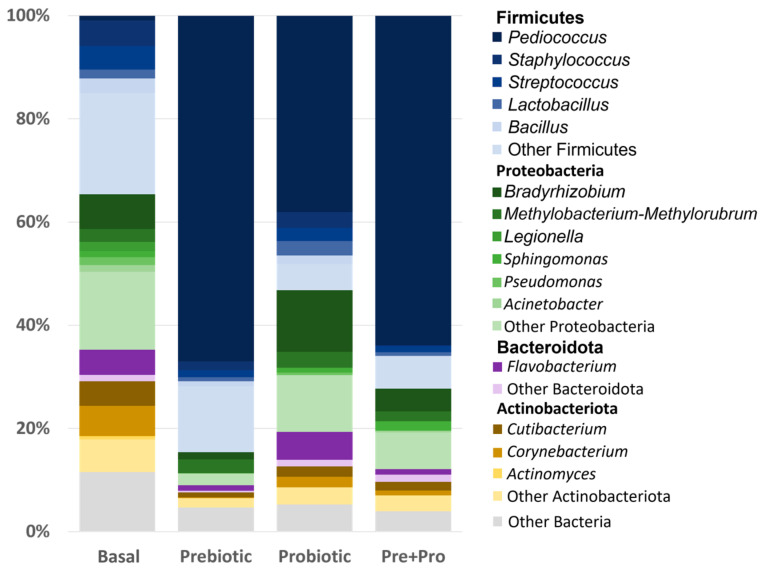
Average relative abundance of bacterial genera in red drum digesta. All samples were found to be predominantly composed of bacteria from four phyla: Firmicutes, Proteobacteria, Bacteroidota, and Actinobacteriota.

**Figure 5 animals-12-02629-f005:**
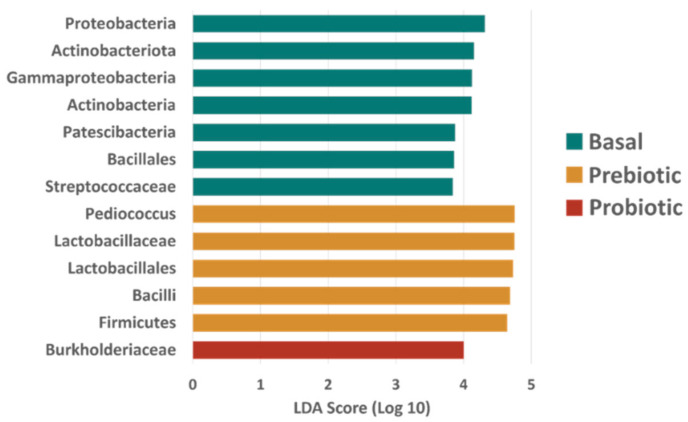
Results of differential taxa abundance testing from Linear Discriminant Analysis (LDA) Effect Size (LEfSe). Taxa shown were found to have significantly higher relative abundance in the specified group relative to all other groups with an LDA Score (log 10) > 3.0 and *p* < 0.05.

**Figure 6 animals-12-02629-f006:**
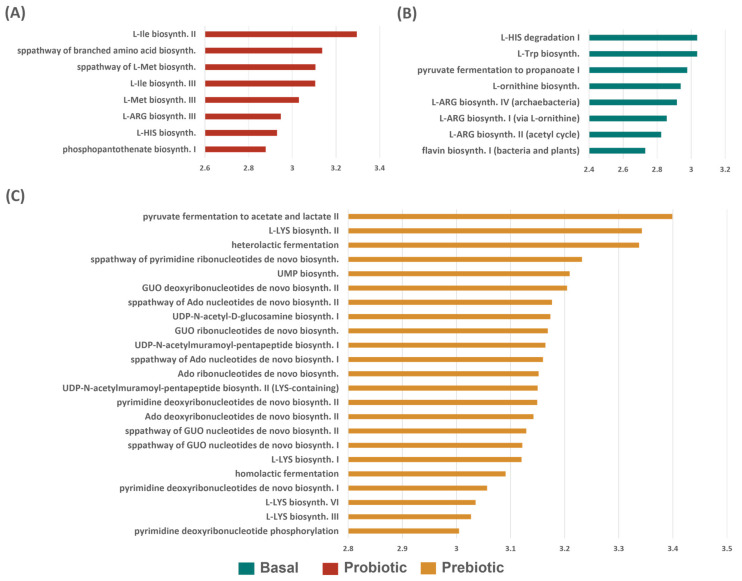
Selected results of differential functional pathway abundance testing from Linear Discriminant Analysis (LDA) Effect Size (LEfSe) grouped by dietary treatment: (**A**) Probiotic, (**B**) Basal, and (**C**) Prebiotic. Functional output pathways of the bacterial microbiota were predicted via PICRUSt2. Pathways shown were found to have significantly higher relative abundance in the specified group relative to all other groups with an LDA Score (log 10) > 2.5 and *p* < 0.05.

**Table 1 animals-12-02629-t001:** Formulations and analyzed proximate composition of the experimental diets. Values are expressed as g/1000 g of diet on a dry-matter basis.

Ingredients	Basal	Prebiotic	Probiotic	Pre + Pro
Menhaden Fishmeal ^1^	154.5	154.5	154.5	154.5
Poultry By-Product Meal ^2^	60.5	60.5	60.5	60.5
Soy Protein Concentrate ^3^	163.0	163.0	163.0	163.0
Dehulled Soybean Meal ^4^	301.0	301.0	301.0	301.0
Dextrinized Corn Starch ^5^	50.0	50.0	50.0	50.0
Fish oil ^1^	84.0	84.0	84.0	84.0
Glycine ^5^	10.0	3.0	10.0	3.0
Lysine ^6^	10.0	10.0	10.0	10.0
Taurine ^5^	10.0	10.0	10.0	10.0
Methionine ^7^	7.5	7.5	7.5	7.5
Carboxymethyl Cellulose ^5^	20.0	20.0	20.0	20.0
Cellufil ^5^	59.5	46.5	58.5	45.5
Grobiotic^®^-A ^8^	0.0	20.0	0.0	20.0
Bactocell^® 9^	0.0	0.0	1.0	1.0
Mineral Premix ^5^	40.0	40.0	40.0	40.0
Vitamin Premix ^10^	30.0	30.0	30.0	30.0
Proximate Composition ^11^				
Dry Matter	903.3	905	903.9	905.2
Protein	471.2	464.8	471.1	470.7
Lipid	137.1	127.5	132.1	126.3
Ash	107.9	110.2	109.3	108.7

Abbreviations: Pre + Pro: Prebiotic and Probiotic; ^1^ Omega Protein Corporation, Abbeville, LO, USA; ^2^ Tyson Foods, Springdale, AR, USA; ^3^ ProFine F. DuPont Nutrition & Biosciences, New Century, KS, USA; ^4^ Producers Cooperative Association, Bryan, TX, USA; ^5^ MP Biomedicals, Solon, OH, USA; ^6^ ADM Animal Nutrition, Quincy, IL, USA; ^7^ Ajinomoto North America Inc., Itasca, IL, USA; ^8^ International Ingredients Corporation, St. Louis, MO, USA; ^9^ Lallemand Animal Nutrition, Milwaukee, WI, USA; ^10^ Same as in Moon & Gatlin III (1991); ^11^ Values presented are means of three replicate analyses.

**Table 2 animals-12-02629-t002:** Production performance and condition indices of red drum fed probiotic, prebiotic, and their combination for 8 weeks.

	Initial Weight (g)	Weight Gain (%)	Feed Efficiency	HSI (%)	IPF (%)	Survival (%)
*Treatment means* ^1^						
Basal	5.59	663.2	0.81	1.59	0.73	95.8
Prebiotic	5.58	618.6	0.74	1.63	0.60	91.6
Probiotic	5.61	769.4	0.81	1.67	0.69	87.5
Pre + Pro	5.59	632.8	0.79	1.74	0.67	97.9
*Main effect means* ^2^						
Prebiotic						
0 g kg^−1^		716.3 ^A^	0.81	1.63	0.71	91.6
20 g kg^−1^		625.7 ^B^	0.76	1.69	0.64	94.8
Probiotic						
0 g kg^−1^		640.9 ^B^	0.77	1.62	0.66	93.7
1 g kg^−1^		701.1 ^A^	0.8	1.71	0.68	92.7
PSE	0.06	26.1	0.02	0.08	0.08	4.29
*Two-way*						
*ANOVA p values*						
Prebiotic		0.007	0.057	0.51	0.41	0.48
Probiotic		0.046	0.18	0.30	0.84	0.81
Pre × Pro		0.11	0.27	0.81	0.49	0.12
Tank Block		0.08	0.88	0.23	0.28	0.43

Abbreviations: HSI: Hepatosomatic index; IPF: Intraperitoneal fat; Pre + Pro: Prebiotic and Probiotic; PSE: Pooled Standard Error. Different superscript letters are significantly different (*p* < 0.05). ^1^ Values represent means of four replicate tanks (*n* = 4). ^2^ Values represent means of eight replicate tanks (*n* = 8).

**Table 3 animals-12-02629-t003:** Immunological responses in the red drum plasma, whole-body proximate composition, and protein conversion efficiency (PCE) after fish were fed the experimental diets for 8 weeks.

	Lysozyme	Total Protein	Total Ig	Moisture	Protein	Lipid	Ash	PCE
	U mL^−1^	mg mL^−1^	mg mL^−1^	% of Whole-Body Tissue on Wet-Basis				%
*Treatment means* ^1^								
Basal	102.0	24.2	13.5	71.9	17.8	15.0	3.9	30.3
Prebiotic	109.4	23.5	13.3	72.5	17.7	15.0	3.8	33.3
Probiotic	108.3	26.9	15.5	70.2	18.6	15.9	3.8	32.6
Pre + Pro	120.8	23.4	12.6	71.4	18.0	17.1	3.9	37.9
*Main effect means* ^2^								
Prebiotic								
0 g kg^−1^	105.2	25.6 ^A^	14.5	71.1	18.2	15.5	3.8	31.4
20 g kg^−1^	115.1	23.5 ^B^	12.9	71.9	17.9	16.1	3.8	35.6
Probiotic								
0 g kg^−1^	105.7	23.9	13.4	72.2	17.8 ^B^	15	3.8	31.8
1 g kg^−1^	114.6	25.2	14	70.8	18.3 ^A^	16.5	3.8	35.2
PSE	18.1	0.9	0.7	0.6	0.6	0.8	0.07	2.34
*Two-way*								
*ANOVA p values*								
Prebiotic	0.59	0.047	0.07	0.22	0.09	0.11	0.85	0.10
Probiotic	0.63	0.18	0.42	0.06	0.007	0.47	0.92	0.17
Pre × Pro	0.88	0.15	0.09	0.65	0.12	0.45	0.36	0.62
Tank Block	0.45	0.21	0.69	0.08	0.02	0.08	0.79	0.14

Abbreviations: Ig: Immunoglobulin; PCE: Protein Conversion Efficiency; Pre + Pro: Prebiotic and Probiotic; PSE: Pooled Standard Error. ^1^ Values represent means of four replicate tanks (*n* = 4). ^2^ Values represent means of eight replicate tanks (*n* = 8).

## Data Availability

Data for the feeding trial can be provided upon reasonable request to the corresponding authors. Next-generation sequencing data is available at the NCBI Sequence Read Archive under the BioProject ID PRJNA 736988.

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
