# Peer review of "Dietary Supplementation of a Commercial Prebiotic, Probiotic and Their Combination Affected Growth Performance and Transient Intestinal Microbiota of Red Drum (Sciaenops ocellatus L.)"

_animals, 2022, doi:10.3390/ani12192629_

Round 1
Reviewer 1 Report
This manuscript entitled “Dietary supplementation of a commercial prebiotic, probiotic and their combination affected production performance and intestinal microbiota of red drum (Sciaenops ocellatus L.) but did not modulate plasma innate immune responses”, describes investigation on the effect of commercial dietry prebiotic and probiotic supplements and their synbiotic effects on growth, immunological parameters and intestinal microbiota of a fast growing marine species, cultured in USA, Red drum.
The study covering good range of factors has never been performed before in Red drum and thus successfully provided an overview of using probiotic and prebiotic and combination of that in a diet of carnivores species which is cultured with low levels of fish-drived ingredient.
Major correction
There is no major corrections on this Manuscript.
Minor correction
Title: production performance is better to replace with growth performance
Abstract: Nothing mentioned in Abstract about blood sampling and plasma immunological assays.
-"In the present study, the potential synergism between beneficial lactic acid bacteria (Pedi- 28 ococcus acidilactici) contained in a probiotic and a mixture of fermentable complex carbohydrates 29 and autolyzed brewer’s yeast (or prebiotic) were explored in red drum." add low fishmeal based diet before red drum.
The last line of abstract, the individual inclusion of these prebiotic and probiotic supplements positively affected the growth performance and intestinal health of red drum, need to change based on your data to the individual inclusion of these probiotic and prebiotic supplements positively affected the growth performance and intestinal health of red drum, respectively. since prebiotic fed fish showed weight gain reduction.
Key words, Sciaenops ocellatus is in the title. need to be different from title.
Introduction-
The Manuscript would benefit from further editing for typographic errors, I have not provided details for topographic errors.
For instance in line seven of introduction the sentence start with And.
References 5 and 6 are not matched with the text. In the text you mentioned several studies but you cited a review paper. Please cite original papers.
Based on your introduction is not clear why you used PCR or denaturing gradient gel electrophoresis or high-throughput sequencing in your study. You need to add a paragraph and explain why these methods has been used in this study and mention the related literature so far for each of the methods.
Material and methods:
What is the colony-forming unit for probiotic diet. Could you provide a target dosage of probiotic in the diet. The diet part need more clarification. Is it extrude diet, why is not sprayed on the diet.
It is highly recommended to separate Experimental feeds from culturing conditions.
Please add a section and explain how you measure proximate composition analyses and PCE in material and method.
In section 2.3 please provide more information about PCR reaction. timing, cycle and reagents.
Results-
Table 2-
Survival in Probiotic group was lower than other group, however, the growth in fish received Probiotic diet was higher compared to other group, is it because of diet or because of lower stocking density.
-For immune assay are you measured it in plasma or serum?
-Figure 6 can move from the text to the supplementary information.
Conclusion
Could you please make it clear what you means by improving production performance.
Author Response
REVIEWER #1
This manuscript entitled “Dietary supplementation of a commercial prebiotic, probiotic and their combination affected production performance and intestinal microbiota of red drum (Sciaenops ocellatus L.) but did not modulate plasma innate immune responses”, describes investigation on the effect of commercial dietry prebiotic and probiotic supplements and their synbiotic effects on growth, immunological parameters and intestinal microbiota of a fast growing marine species, cultured in USA, Red drum.
The study covering good range of factors has never been performed before in Red drum and thus successfully provided an overview of using probiotic and prebiotic and combination of that in a diet of carnivores species which is cultured with low levels of fish-drived ingredient.
Thank you for your comments and for reviewing our manuscript. We hope to have addressed all your concerns and we remain at your disposal for further clarification.
Major correction
There is no major corrections on this Manuscript.
Minor correction
Title: production performance is better to replace with growth performance
Thank you for your suggestion. Modifications were made accordingly.
Abstract: Nothing mentioned in Abstract about blood sampling and plasma immunological assays.
-"In the present study, the potential synergism between beneficial lactic acid bacteria (Pedi- 28 ococcus acidilactici) contained in a probiotic and a mixture of fermentable complex carbohydrates 29 and autolyzed brewer’s yeast (or prebiotic) were explored in red drum." add low fishmeal based diet before red drum.
The last line of abstract, the individual inclusion of these prebiotic and probiotic supplements positively affected the growth performance and intestinal health of red drum, need to change based on your data to the individual inclusion of these probiotic and prebiotic supplements positively affected the growth performance and intestinal health of red drum, respectively. since prebiotic fed fish showed weight gain reduction.
The maximum word count on the abstract (200 words) limited the information we could provide. We decided to highlight the findings that were significantly improved.
Key words, Sciaenops ocellatus is in the title. need to be different from title.
Thank you for your suggestion. Modifications were made accordingly.
Introduction-
The Manuscript would benefit from further editing for typographic errors, I have not provided details for topographic errors.
For instance in line seven of introduction the sentence start with And.
Thank you for your suggestion. Modifications were made accordingly.
References 5 and 6 are not matched with the text. In the text you mentioned several studies but you cited a review paper. Please cite original papers.
Thank you for your suggestion. The authors presented information from the review papers where several studies presented positive results testing these commercial products. The original papers are then cited in the discussion.
Based on your introduction is not clear why you used PCR or denaturing gradient gel electrophoresis or high-throughput sequencing in your study. You need to add a paragraph and explain why these methods has been used in this study and mention the related literature so far for each of the methods.
Material and methods:
What is the colony-forming unit for probiotic diet. Could you provide a target dosage of probiotic in the diet. The diet part need more clarification. Is it extrude diet, why is not sprayed on the diet.
It is highly recommended to separate Experimental feeds from culturing conditions.
Please add a section and explain how you measure proximate composition analyses and PCE in material and method.
Thank you for your suggestion. Authors tried to refrain from repeating information in the material and methods, and referenced a study detailing the feed manufacture established procedures. This information can be found in L99-101 from the original submission. Information on PCE can be also found in the formula L170 from the original submission.
In section 2.3 please provide more information about PCR reaction. timing, cycle and reagents.
Thank you for your suggestion. In order to avoid repeating the same methods, the authors prefer to reference a manuscript where the procedures are detailed.
Results-
Table 2-
Survival in Probiotic group was lower than other group, however, the growth in fish received Probiotic diet was higher compared to other group, is it because of diet or because of lower stocking density.
Survival was not significantly affected by the dietary treatment, and it was not homogeneously spread across the experimental units. Thus, we did not consider the lower stocking density a factor for the increased growth performance.
-For immune assay are you measured it in plasma or serum?
Plasma. L216 from the original manuscript provide this information.
-Figure 6 can move from the text to the supplementary information.
Thank you for your suggestion. A whole paragraph in the discussion is devoted for Figure 6. The authors prefer to keep the figure in the article.
Conclusion
Could you please make it clear what you means by improving production performance.
Thank you for your suggestion. The sentence was reworded for growth performance

Reviewer 2 Report
Yamamoto et al. explored the effects of one prebiotic (Grobiotic-A), one probiotic (Bactocell) and its combination over the fish performance and microbiome in red drum (Sciaenops ocellatus). How fish nutrition affects intestinal microbiome is a hot topic nowadays considering the important impact that different compounds might have in the microbial communities, and how this determines the health status of the host organism and the potential immune response when facing bacterial diseases. In general, I regret to consider that the present manuscript is not suitable for publication in Animals journal, at least on its present form. This is not considered by the lack of the novelty, as some previous reports on the use of the same probiotics and prebiotics have been already published, this is exclusively based on the following major issues that authors need to address listed below.
Major issues:
1-Authors do not present adequately the results obtained from the present study. Authors did not include SD values for each evaluated parameter on each experimental group.
2-Authors conclude that the immune response is not different between experimental groups. However, authors did not evaluate the “response” of the immune system by a pathogenic challenge.
3-Material and methods. Some important descriptions on particular analyses are not included. This is mandatory for experimental replication if other researchers considered necessarily and/or interesting.
4-Resuls by DGGE and DNA-Seq are not consistent. Infact, results are contradictory.
5-Results presented in tables and/or figures are not including the values from each biological replicate from each group.
6-Conclusion regarding how Illumina platform MiSeq is hampering the ability to identify particular species seems to be inaccurate, suggesting that authors are not very familiar with this experimental approach and/or analytics.
Minor details:
L2: and/or
L4: regarding the immune response. This has been not properly addressed. Authors need to perform a bacterial challenge in order to conclude that an immune response has been modulated. In this sense, an immune response is only triggered when facing a disease. Here no challenge has been done. Please, rephrase.
L21: scientific name?
L30: scientific name?
L35-40: Authors should include some description about the use of the prebiotic.
L43: Please, do not use as keywords terms that have been used in the title (Sciaenops ocellatus).
L55-56: the single use
L72-76: I think that authors should clarify if the purpose was to provide a prebiotic for improving the colonization of lactic acid bacteria? Or just other kind of microorganisms?
L86-87: Since only one supplementation of both products is tested, the introduction lacks the rationale of how this concentration has been selected. It is the same as tested in other fish species?
L93-94: which was the level of supplementation? In table 1 is indicated 20 g/kg. But which is the rationale behind?
L131-132: Please, indicate the Lux at Surface.
L136-139: How many fish were sampled for the initial nutrient composition? Also, why whole-body composition was analyzed instead of fillet composition? The main point is the nutritional composition of what consumers eat. In addition, including several tissues, there is a chance that potential differences might be masked.
L141-142: differences in feed intake?
L159: why fishes were not weighted individually? This might provide an insight on the variability within each tank.
L164: no viscerosomatic index was evaluated?
L172: Viscerosomatic? The included formula is for hepatosomatic index
L177-178: why authors are only including 12 fish per tank if they will sample 9 fishes. The decreased fish density at the end might have some effect on the microbiome?
L182: Please, avoid self-citations. Yamamoto is not a technical paper where an analytical procedure is particularly described!
L195: the region amplified is V3-V4? Here V3 is indicated, but after (L202) V4 is used.
L202-209: Please, describe the protocols for bacterial DNA isolation and libraries sequencing.
L204: 30 cycles of amplification is the number of cycles suggested to be used in the corresponding protocol?
L204: in addition to the bioproject, authors should present a table compiling the sequencing output including: number of raw reads, trimmed reads, GC content, sequencing quality scores, etc.... for each sample. How many samples were individually sequenced per experimental group?
L211-212: Base on the functional analysis in human or mammals?
L214: authors have confirmed that blood samples do not include some hemolysis?
L217-219: Bradford assay?
L220: Please, indicate the equipment. Also, how specific and accurate is the method for quantifying immunoblobulins here used? Please, discus it and indicate accuracy when comparing to western blot.
L228-231: Why not conducting Tukey if no interaction was found?
L244-245: why two different criteria for significance were considered?
L253-256: how increased WB protein content was observed without differences in protein conversion efficiency? Any clue?
L257: how plasma lysozyme activity was evaluated was not described in mat and methods.
L258-259: How higher plasma total protein concentration in fish fed prebiotic might be correlated with the reduced growth in prebiotic group? Please, discuss this.
L261: Please, for all the parameters evaluated the mean and SD values. Also, please, include final weight values in the table.
Table 1: Final weight gain of probiotic group was higher, but showing lower survival rate. Feed addition was corrected considering this lower fish survival?
L263: Pooled standard error? In materials and methods, it was indicated that results are presented as mean values ± SD. Why here it is SE? In addition, SD representation provides a more intuitive idea of how values from replicates varies with respect to mean value. Please, present SD values for each experimental group.
Table 2: also, why in some cases values presented are means of 8 biological replicates. Only 4 were performed. Even when no interaction between the two factors considered (probiotic and prebiotic addition) are not interacting, they cannot be considered as not have an effect, and thus no assumption regarding they might be replicates can be done.
L267: why whole-body composition was analyzed? Fish fillet composition would be better, as no different tissues are included in the analysis that might mask potential differences.
Table 3: Same comment as in table 2
Table 3: What means that Tank bloks is significantly different (P=0.02)?
Figure 1: Only one sample per experimental group has been represented. Please, include the results from the 4 biological replicates (tanks, n=4) or the whole technical replicates, that is the 12 (3 pseudo-replicates from each tank; 3 x 4 =12) replicates.
In addition, the percentage of similarity between groups is 91. This means that all samples are very similar. If NGS analysis has been done, while this analysis (much less accurate and powerful) has been performed? In fact, contradictory results in figure 3 and 4 are shown. Probiotic is more similar to Basal diet.
Figure 2: It seems that any experimental group is different from the Control (Basal) one. This is not consistent with the results from the dendrogram. Please, indicate what is represented with the boxes of the boxplot, and the horizontal line in the middle of each box. Also, indicate the number of biological replicates analyzed for each group.
Figure 3: Please, indicate what is represented by each dot. The mean average of the 3 technical replicates analyzed within each tank, or the pooled sampling of these 3 technical replicates? If R= 0.5 or 0.39, what is the biological significance of having this low correlation coefficient. Please, discuss this. Also, authors perform a PCoA analysis, but which are the families/genus that mostly explain the differences/variability between experimental groups?
L302: pre and probiotic?
Figure 4: The bacterial families are here represented? Please, clearly state what is represented in each figure (bacterial families? Genus?). Also, represent the 4 replicates from each group to clearly shown that all replicates are responding equally. A heatmap showing how the replicates from different experimental groups are clustering is required.
In addition, what is the significance that flavobacterium relative abundance is similar between Basal and Probiotic groups?
Figure 5: families? Authors should present a heat map of the different biological replicates and identify the families with different relative abundance. This figure not including mean values and SD is not informative and/or intuitive as it should be.
L323 and Figure 6: why only pathways of nutritional significance as displayed? They are statistically differentially represented? Any other pathway differentially represented should be represented and discussed. Also, Mean and SD deviation should be presented.
L329-330: Why no significant pathways were observed for the Pre-Pro group if there is substantial differences in the relative abundance of some bacterial families? What about the results on genus?
L360: how high was the inclusion of soybean? Please, compare the levels of inclusion/substitution with the ones here tested.
L367-370: it has been scientifically reported? Please, include the reference.
L370-372: if not providing any article supporting this statement, e.g. any article that has analyzed the activity of antinutritional factors or the activity of protease (e.g. trypsin), this is a mere speculation. Please, avoid it.
L375-377: this is right, but no challenge has been done to demonstrate that the immune response is different (or not) in any experimental group upon bacterial infection.
L378: again, no immune response has been evaluated.
L381-392: considering the knowledge available regarding the effects of Bactocell over the immune system in different fish species. Why authors did not explore the already potential biomarkers (e.g. IL-6 and 10)? Only lysozyme and total plasma immunoglobulins were assessed, which are not considered as the best (more robust and accurate) biomarker of fish immune system.
L393-395: The present study should not rely on this DGGE analysis, as it is not further supported by a more accurate and powerful analysis (DNA-Seq).
L405: bacterial taxa is not very specific. Family? Genus?
L406-408: This assumption seems to be not considering dysbiosis. Not always higher diversity means better health status. For instance, if authors represent bacterial composition at genus level, perhaps some of the genus at the basal diet are secondary/opportunistic pathogens. Authors should analyze the bacterial composition at genus level and explore which genus are present/abundant in the different groups.
L413-416: in addition to this hypothesis, what about the inhibitory effect over other bacteria?
L421: these results of the PCoA are not supporting the results with the DGGE. Any clue?
L424-427: this has not been shown. Please, show the bacterial composition at genus level.
L427-430: this limitation is not due to Illumina MiSeq. It might be due to the primers and the genomic region considered to evaluate the bacterial composition. A specific amplification of the species specific Pediococcus specific sequence might shed some light on this issue.
L450-453: No other reports can corroborate present findings. It can suggest, but never corroborate.
L466: This has not been demonstrated here. This can be said about the probiotic group, but no regarding the other experimental groups. Please, rephrase it.
L469: Only pathogenic species?
Supplementary figure: Please, delete it. Supplementary figures should be not remainders.
Author Response
REVIEWER #2
Yamamoto et al. explored the effects of one prebiotic (Grobiotic-A), one probiotic (Bactocell) and its combination over the fish performance and microbiome in red drum (Sciaenops ocellatus). How fish nutrition affects intestinal microbiome is a hot topic nowadays considering the important impact that different compounds might have in the microbial communities, and how this determines the health status of the host organism and the potential immune response when facing bacterial diseases. In general, I regret to consider that the present manuscript is not suitable for publication in Animals journal, at least on its present form. This is not considered by the lack of the novelty, as some previous reports on the use of the same probiotics and prebiotics have been already published, this is exclusively based on the following major issues that authors need to address listed below.
Thank you for your comments and for reviewing our manuscript. We hope to have addressed all your concerns and we remain at your disposal for further clarification.
Major issues:
1-Authors do not present adequately the results obtained from the present study. Authors did not include SD values for each evaluated parameter on each experimental group.
The authors would like to keep the presentation of their data as is. There is no specific instructions in the authors’ guidelines that the average should be followed by the standard deviation. Moreover, two-way ANOVA tables are often presented this way because they already have a lot of information and adding the standard deviation to every mean would make the tables unnecessarily larger.
Some examples of other published studies using the same format can be found below:
Li, M.H.; Robinson, E.H.; Tucker, C.S.; Oberle, D.F.; Bosworth, B.G. Comparison of Channel Catfish, Ictalurus punctatus, and Blue Catfish, Ictalurus furcatus, Fed Diets Containing Various Levels of Protein in Production Ponds. J. World Aquac. Soc. 2008, 39, 646–655, doi:10.1111/j.1749-7345.2008.00200.x.
Robinson, E.H.; Li, M.H. Effect of Dietary Protein Concentration and Feeding Rate on Weight Gain, Feed Efficiency, and Body Composition of Pond-Raised Channel Catfish Ictalurus punctatus. J. World Aquac. Soc. 1999, 30, 311–318, doi:10.1111/j.1749-7345.1999.tb00681.x.
Buentello, J.A.; Gatlin, D.M. The Dietary Arginine Requirement of Channel Catfish (Ictalurus punctatus) Is Influenced by Endogenous Synthesis of Arginine from Glutamic Acid. Aquaculture 2000, 188, 311–321, doi:10.1016/S0044-8486(00)00344-6.
Yildirim‐Aksoy, M.; Lim, C.; Li, M.H.; Klesius, P.H. Interaction between Dietary Levels of Vitamins C and E on Growth and Immune Responses in Channel Catfish, Ictalurus punctatus (Rafinesque). Aquac. Res. 2008, 39, 1198–1209.
Barros, M.M.; Lim, C.; Klesius, P.H. Effect of Soybean Meal Replacement by Cottonseed Meal and Iron Supplementation on Growth, Immune Response and Resistance of Channel Catfish (Ictalurus puctatus) to Edwardsiella ictaluri Challenge. Aquaculture 2002, 207, 263–279.
2-Authors conclude that the immune response is not different between experimental groups. However, authors did not evaluate the “response” of the immune system by a pathogenic challenge.
Despite not having an experimental pathogenic challenge, which the authors acknowledge it was a limitation of the study; however, it does not invalidate it. Numerous studies have used similar methods to those described in the presented manuscript to evaluate immune response without a bacterial challenge:
Mohammadi Arani, M.; Salati, A.P.; Safari, O.; Keyvanshokooh, S. Dietary Supplementation Effects of Pediococcus acidilactici as Probiotic on Growth Performance, Digestive Enzyme Activities and Immunity Response in Zebrafish (Danio rerio). Aquac. Nutr. 2019, 25, 854–861, doi:10.1111/anu.12904.Ashouri, G.; Mahboobi Soofiani, N.; Hoseinifar, S.H.; Jalali, S.A.H.; Morshedi, V.; Van Doan, H.; Torfi Mozanzadeh, M. Combined Effects of Dietary Low Molecular Weight Sodium Alginate and Pediococcus acidilactici MA18/5M on Growth Performance, Haematological and Innate Immune Responses of Asian Sea Bass (Lates calcalifer) Juveniles. Fish Shellfish Immunol. 2018, 79, 34–41, doi:10.1016/j.fsi.2018.05.009.
González-Félix, M.L.; Gatlin, D.M.; Urquidez-Bejarano, P.; de la Reé-Rodríguez, C.; Duarte-Rodríguez, L.; Sánchez, F.; Casas-Reyes, A.; Yamamoto, F.Y.; Ochoa-Leyva, A.; Perez-Velazquez, M. Effects of Commercial Dietary Prebiotic and Probiotic Supplements on Growth, Innate Immune Responses, and Intestinal Microbiota and Histology of Totoaba macdonaldi. Aquaculture 2018, 491, 239–251, doi:10.1016/j.aquaculture.2018.03.031.
Zhou, Q.C.; Buentello, J.A.; Gatlin, D.M. Effects of Dietary Prebiotics on Growth Performance, Immune Response and Intestinal Morphology of Red Drum (Sciaenops ocellatus). Aquaculture 2010, 309, 253–257, doi:10.1016/j.aquaculture.2010.09.003.
In addition, in L422-424, we indicate that further research involving bacterial challenge or stressful conditions should be performed.
3-Material and methods. Some important descriptions on particular analyses are not included. This is mandatory for experimental replication if other researchers considered necessarily and/or interesting.
Additional information was added as requested.
4-Resuls by DGGE and DNA-Seq are not consistent. Infact, results are contradictory.
The DGGE is a qualitative method that allows for characterization of the bacterial community based on the different structures of PCR amplicons using visualization of communities with gel electrophoresis. Using the Illumina MiSeq, we sequence the bacterial amplicons, allowing for identification of the specific taxa using an established database. The output of this process can be quantified and analyzed using an array of metrics, as presented in our study. These two methods therefore inherently have differences in sensitivity and uses, with DGGE giving a broad, but limited, overview of community diversity and NGS allowing for a more specific overview of community diversity, via identification of taxa, and allows for more quantitative diversity analyses.
5-Results presented in tables and/or figures are not including the values from each biological replicate from each group.
Where appropriate, figures do incorporate values from biological replicates, for example figures 2 and 3. However this information does not always easily fit into tables or figures given the number of samples, and therefore the data are summarized within groups which is not only stated in the methods and legends, but is also a common method of data presentation.
6-Conclusion regarding how Illumina platform MiSeq is hampering the ability to identify particular species seems to be inaccurate, suggesting that authors are not very familiar with this experimental approach and/or analytics.
Although species level identifications can be obtained using some classification methods for Illumina MiSeq data, these are not always accurate. Illumina MiSeq can provide high-quality reads in a short length (~300 bp). If the forward and reverse reads are merged, they could account for up to ~500 bp, depending on the region amplified and sequenced. However, the whole 16S rRNA gene sequence length is ~1500 bp, and it has been previously shown that short-read sequencing of variable regions of the 16S rRNA gene cannot provide accurate species-level resolution (Johnson et al., 2019). More information on this subject can be found in the following articles:
Johnson, J.S., et al. Evaluation of 16S rRNA gene sequencing for species and strain-level microbiome analysis. Nat Commun, 2019. 10(1): p. 5029.
Nygaard et al., etal. A preliminary study on the potential of Nanopore MinION and Illumina MiSeq 16S rRNA gene sequencing to characterize building-dust microbiomes. Sci Rep, 2020.
Minor details:
L2: and/or
L4: regarding the immune response. This has been not properly addressed. Authors need to perform a bacterial challenge in order to conclude that an immune response has been modulated. In this sense, an immune response is only triggered when facing a disease. Here no challenge has been done. Please, rephrase.
Thank you for your suggestion. Modifications were made accordingly and the concern with bacterial challenge has been addressed above.
L21: scientific name?
L30: scientific name?
The simple summary is meant to deliver the scientific information for a lay audience. Thus, scientific names were not included in this segment of the manuscript.
L35-40: Authors should include some description about the use of the prebiotic.
The word count for the abstract limited the information that we could include. Thus it was reserved mainly for the significant findings of the study.
L43: Please, do not use as keywords terms that have been used in the title (Sciaenops ocellatus).
Thank you for your suggestion. Modifications were made accordingly.
L55-56: the single use
The authors don’t understand what the reviewer is asking.
L72-76: I think that authors should clarify if the purpose was to provide a prebiotic for improving the colonization of lactic acid bacteria? Or just other kind of microorganisms?
The information was provided at L59-61 from the original manuscript.
L86-87: Since only one supplementation of both products is tested, the introduction lacks the rationale of how this concentration has been selected. It is the same as tested in other fish species?
L93-94: which was the level of supplementation? In table 1 is indicated 20 g/kg. But which is the rationale behind?
Thank you for your question. This information was added to the material and methods section L95-96 for the R1.
L131-132: Please, indicate the Lux at Surface.
Thank you for your suggestion. Unfortunately, we do not have this information.
L136-139: How many fish were sampled for the initial nutrient composition? Also, why whole-body composition was analyzed instead of fillet composition? The main point is the nutritional composition of what consumers eat. In addition, including several tissues, there is a chance that potential differences might be masked.
Thank you for your question. The number of fish was not counted, as we used the biomass as the main parameter to ensure we would have enough material for proximate analysis. Indeed, the main consumer is the end point, and more research should be performed with food size fish. However, at the end of the study, fish were ~50 grams each, and this is ~1/30 of the harvest size; and therefore, would not be a good representation of what is currently being employed by the industry.
L141-142: differences in feed intake?
L159: why fishes were not weighted individually? This might provide an insight on the variability within each tank.
In order to avoid over stressing the animals, fish were group weighed. This procedure may be feasible with tilapia, or catfish, but in our experience, handling the animals too much would increase their susceptibility to pathogens.
L164: no viscerosomatic index was evaluated?
L172: Viscerosomatic? The included formula is for hepatosomatic index
The viscerosomatic indices evaluated were IPF or HSI.
L177-178: why authors are only including 12 fish per tank if they will sample 9 fishes. The decreased fish density at the end might have some effect on the microbiome?
Thank you for your input. The limited volume of the 38-L tank only allowed us to stock 12 fish per tank. In addition, if the decrease in the stocking density has an impact on the microbiome, it would be equally distributed among all dietary treatments.
L182: Please, avoid self-citations. Yamamoto is not a technical paper where an analytical procedure is particularly described!
Although unnecessary self-citations should be avoided, here this paper is referenced since the procedures that are explicitly described in the referenced article were utilized in the present manuscript.
L195: the region amplified is V3-V4? Here V3 is indicated, but after (L202) V4 is used.
The DGGE used the V3 region, and the Illumina MiSeq sequenced amplicons from the V4 region.
L202-209: Please, describe the protocols for bacterial DNA isolation and libraries sequencing.
The protocol for the bacterial DNA extraction can be found on L188-189 from the original submission. Protocol for library preparation and sequencing can be found in the reference Gohl et al [16] (L204).
L204: 30 cycles of amplification is the number of cycles suggested to be used in the corresponding protocol?
The referenced study which describes the protocol used for library preparation does not suggest a specific number of PCR cycles since this depends on the sample type. The manuscript does state that “…the number of PCR cycles should be minimized” and the sequencing facility, which is led by some of the authors of the referenced study, had suggested 30 cycles based on their initial sample processing steps and expertise.
L204: in addition to the bioproject, authors should present a table compiling the sequencing output including: number of raw reads, trimmed reads, GC content, sequencing quality scores, etc.... for each sample. How many samples were individually sequenced per experimental group?
This information was not added as the authors feel that the article is lengthy as is and this information would not add much value. The information included in the present manuscript is appropriate for this technique.
L211-212: Base on the functional analysis in human or mammals?
Based on the bacterial metagenome. More information on this analysis can be found in Douglas et al. PICRUSt2 for Prediction of Metagenome Functions. Nat. Biotechnol. 2020, 1–5.
L214: authors have confirmed that blood samples do not include some hemolysis?
Blood samples were immediately centrifuged after being collected. No hemolysis happened in this time frame.
L217-219: Bradford assay?
L220: Please, indicate the equipment. Also, how specific and accurate is the method for quantifying immunoblobulins here used? Please, discus it and indicate accuracy when comparing to western blot.
Western blot does not provide us a concentration of immunoglobulins in the plasma. If a more advanced analyses could be done for this assay it would be a sandwich ELISA. But for that we will need a monoclonal antibodies against red drum antibodies. At the time, we did not have the resources to develop monoclonal antibodies.
L228-231: Why not conducting Tukey if no interaction was found?
Student T-test is more appropriate when testing for two different treatments.
L244-245: why two different criteria for significance were considered?
Although using a LDA score (log 10)>2.5 would be appropriate for both analyses, our analyses resulted in many taxa which were considered significant, a higher cut-off (more stringent) was chosen for the taxa given the number of significant findings, which would have resulted in an overly busy figure 5.
L253-256: how increased WB protein content was observed without differences in protein conversion efficiency? Any clue?
As the formula stated for the PCE, there are variables that can increase the variance of the results, precluding the ability of the statistical analysis to detect significant differences.
L257: how plasma lysozyme activity was evaluated was not described in mat and methods.
Thank you for your comment. A sentence was added on the material and methods for lysozyme.
L258-259: How higher plasma total protein concentration in fish fed prebiotic might be correlated with the reduced growth in prebiotic group? Please, discuss this.
This was an unknown phenomenon that the authors refrained to speculate.
L261: Please, for all the parameters evaluated the mean and SD values. Also, please, include final weight values in the table.
The authors prefer to keep the tables as is. There is no specific presentation guideline from Animals requiring standard deviations.
Table 1: Final weight gain of probiotic group was higher, but showing lower survival rate. Feed addition was corrected considering this lower fish survival?
Biomass lost for mortality was accounted for in the final calculations.
L263: Pooled standard error? In materials and methods, it was indicated that results are presented as mean values ± SD. Why here it is SE? In addition, SD representation provides a more intuitive idea of how values from replicates varies with respect to mean value. Please, present SD values for each experimental group.
In the materials & methods, “(average + SD)”: immediately follows or precedes results within the text that are presented in this manner, but do not indicate this is how all data are presented.
Again, there are no guidelines to present the values as standard deviation. This is a preference of style.
Table 2: also, why in some cases values presented are means of 8 biological replicates. Only 4 were performed. Even when no interaction between the two factors considered (probiotic and prebiotic addition) are not interacting, they cannot be considered as not have an effect, and thus no assumption regarding they might be replicates can be done.
This is how results look like when analyzing for a two-way anova for the main factors. More information can be found in Ott & Longnecker, An Introduction to Statistical Methods and Data Analysis.
L267: why whole-body composition was analyzed? Fish fillet composition would be better, as no different tissues are included in the analysis that might mask potential differences.
Fish were 50 grams of size. Fillets from fish this size do not represent what is actually performed in the industry.
Table 3: Same comment as in table 2
Table 3: What means that Tank bloks is significantly different (P=0.02)?
It means that the statistical block had a significant effect on that parameter.
Figure 1: Only one sample per experimental group has been represented. Please, include the results from the 4 biological replicates (tanks, n=4) or the whole technical replicates, that is the 12 (3 pseudo-replicates from each tank; 3 x 4 =12) replicates.
This study was conducted a few years ago, and unfortunately, the leading scientist does not work at the same university, which would not allow us to perform this analysis again. Moreover, this is a qualitative analysis where the replicates of the DNA samples were pooled, to identify if the absence or presence of DNA bands can indicate differences for the dietary treatments. The methods have been modified to clarify sample numbers.
In addition, the percentage of similarity between groups is 91. This means that all samples are very similar. If NGS analysis has been done, while this analysis (much less accurate and powerful) has been performed? In fact, contradictory results in figure 3 and 4 are shown. Probiotic is more similar to Basal diet.
Figure 2: It seems that any experimental group is different from the Control (Basal) one. This is not consistent with the results from the dendrogram. Please, indicate what is represented with the boxes of the boxplot, and the horizontal line in the middle of each box. Also, indicate the number of biological replicates analyzed for each group.
Indeed, the results do look different, which was explained above. This is a standard boxplot wherein the horizontal line in the middle of each box indicates the median, boxes include interquartile range, etc. Since this is basic stastitics, we do not feel it is necessary to define what represents a boxplot.
Figure 3: Please, indicate what is represented by each dot. The mean average of the 3 technical replicates analyzed within each tank, or the pooled sampling of these 3 technical replicates? If R= 0.5 or 0.39, what is the biological significance of having this low correlation coefficient. Please, discuss this. Also, authors perform a PCoA analysis, but which are the families/genus that mostly explain the differences/variability between experimental groups?
Within a PCoA, each dot represents a single sample. This is alluded to in the legend of figure 3 with “Samples cluster by”. The biological significance of R values is somewhat subjective – although a 1 indicates dissimilarity and 0 indicates no dissimilarity, there is not a cut off for what would be considered biologically significant. Therefore, we respectfully disagree with discussing the biological significance of these different R values, especially considering the manuscript is already quite long and these results are only briefly mentioned (only in qualitative terms based on visualization with PCoA plots) in the discussion.
The taxa that were significantly affected by the dietary treatments can be found in figure 5.
L302: pre and probiotic?
Thank you for noticing this. This was corrected in the manuscript.
Figure 4: The bacterial families are here represented? Please, clearly state what is represented in each figure (bacterial families? Genus?). Also, represent the 4 replicates from each group to clearly shown that all replicates are responding equally. A heatmap showing how the replicates from different experimental groups are clustering is required.
We have revised the legend to specify these are genera.
The choice of including a plot including either average (as is done here) or individual samples is personal preference. Here, we have chosen to show this data as averages and prefer to keep it that way.
The inclusion of a heatmap is also personal preference and we prefer to keep the data presented as is.
In addition, what is the significance that flavobacterium relative abundance is similar between Basal and Probiotic groups?
There are no significant differences in the relative abundances for Flavobacterium.
Figure 5: families? Authors should present a heat map of the different biological replicates and identify the families with different relative abundance. This figure not including mean values and SD is not informative and/or intuitive as it should be.
This includes any bacterial taxa. As said previously, we disagree with a heatmap being a requirement and prefer to keep the figure as is. This figure indicates which taxa are identified as having significant differential abundance within whichever group the taxa is colored as, with a higher LDA score (log 10) indicating a higher impact on differences between groups. This is explained in the manuscript introducing this method, which is cited for readers in the methods:
Segata, N.; Izard, J.; Waldron, L.; Gevers, D.; Miropolsky, L.; Garrett, W.S.; Huttenhower, C. Metagenomic Biomarker Dis-covery and Explanation. Genome Biol. 2011, doi:10.1186/gb-2011-12-6-r60.
L323 and Figure 6: why only pathways of nutritional significance as displayed? They are statistically differentially represented? Any other pathway differentially represented should be represented and discussed. Also, Mean and SD deviation should be presented.
Since the treatment of interest is diet, the impact on bacterial nutrition was of interest. All other pathways are shown in supplementary figure 1. These are pathways which are found to be significantly differentially represented, which is stated in the legend.
L329-330: Why no significant pathways were observed for the Pre-Pro group if there is substantial differences in the relative abundance of some bacterial families? What about the results on genus?
The reviewer must be confused. Pre+Pro group did not present any significant differences in the discriminant linear analyses (LEFSe) for the bacterial taxa.
L360: how high was the inclusion of soybean? Please, compare the levels of inclusion/substitution with the ones here tested.
Thank you for your suggestion. This information was added in the manuscript.
L367-370: it has been scientifically reported? Please, include the reference.
L370-372: if not providing any article supporting this statement, e.g. any article that has analyzed the activity of antinutritional factors or the activity of protease (e.g. trypsin), this is a mere speculation. Please, avoid it.
The authors prefer to keep the sentences as is.
L375-377: this is right, but no challenge has been done to demonstrate that the immune response is different (or not) in any experimental group upon bacterial infection.
L378: again, no immune response has been evaluated.
This has been addressed above.
L381-392: considering the knowledge available regarding the effects of Bactocell over the immune system in different fish species. Why authors did not explore the already potential biomarkers (e.g. IL-6 and 10)? Only lysozyme and total plasma immunoglobulins were assessed, which are not considered as the best (more robust and accurate) biomarker of fish immune system.
There are many methods/biomarkers which may be useful to further evaluate the fish immune response. We agree there are many that would be interesting to evaluates in further studies and explicitly state throughout the discussion that further research are needed.
L393-395: The present study should not rely on this DGGE analysis, as it is not further supported by a more accurate and powerful analysis (DNA-Seq).
We agree that we should not rely only on DGGE, and therefore we also performed NGS. However we feel it is valuable for readers to see the data resulting from these two different methods.
L405: bacterial taxa is not very specific. Family? Genus?
L406-408: This assumption seems to be not considering dysbiosis. Not always higher diversity means better health status. For instance, if authors represent bacterial composition at genus level, perhaps some of the genus at the basal diet are secondary/opportunistic pathogens. Authors should analyze the bacterial composition at genus level and explore which genus are present/abundant in the different groups.
The next sentences discuss that higher diversity does not always indicate better health status and gives an potential explanation for the observed decreased diversity:
“ However, similar to the present study, a lower diversity was observed for totoaba treated with diets containing moderate inclusion levels of soybean meal and supplemented with the prebiotic Agavin [51]. The high relative abundance of Pediococcus spp., as observed in the relative abundance taxa plot, may be the main driver of the reduced diversity for the intestinal microbiota of prebiotic and Pre+Pro fed red drum. Nevertheless, the reduced evenness observed in fish fed diets containing pre- and/or probiotics may not necessarily have a negative impact on fish health”
Bacterial composition is presented at genus level on figure 4. Relative abundance of all taxonomic levels between phylum and genus were evaluated using the LEfSe analysis.
L413-416: in addition to this hypothesis, what about the inhibitory effect over other bacteria?
This is not within the scope of the manuscript.
L421: these results of the PCoA are not supporting the results with the DGGE. Any clue?
This has already been addressed above. The PCoA results are from a different method targeting a different portion of the 16S rRNA gene.
L424-427: this has not been shown. Please, show the bacterial composition at genus level.
Figure 4 shows the bacterial composition at the genus level.
L427-430: this limitation is not due to Illumina MiSeq. It might be due to the primers and the genomic region considered to evaluate the bacterial composition. A specific amplification of the species specific Pediococcus specific sequence might shed some light on this issue.
This has been addressed above. You are correct, qPCR for various Pediococcus species could be valuable for understanding this but was not the scope of this manuscript. Within the manuscript we have stated that further research including additional methods are needed.
L450-453: No other reports can corroborate present findings. It can suggest, but never corroborate.
L466: This has not been demonstrated here. This can be said about the probiotic group, but no regarding the other experimental groups. Please, rephrase it.
L469: Only pathogenic species?
Supplementary figure: Please, delete it. Supplementary figures should be not remainders.
We respectfully disagree and prefer to keep this figure. Readers may be interested in the results of metagenome functional prediction, other than the ones related to nutrition which are presented in the main text.

Reviewer 3 Report
Major:
The authors need to specify they examined the transient microbiota and not the resident microbiota with the resident microbiota having a greater influence on the host.
Did the authors sample the feed to determine the microbiota present?
Did the authors detect the probiotc in the intestinal samples?
Minor:
427-430: This is a difficult conclusion to make without knowing the composition of the microbiota on the feed before feeding as this will influence the transient microbiome.
441-448: Without knowing the species it is difficult to claim that reduction in the relative abundance of Streptococcus is beneficial to the host. There are several species of Streptococcus that are part of the normal flora of the gut.
Author Response
Major
Thank you for reviewing our manuscript. We hope to have addressed your questions and concerns, and we remain at your disposal for further clarification.
The authors need to specify they examined the transient microbiota and not the resident microbiota with the resident microbiota having a greater influence on the host.
Thank you for your suggestion. Modifications were made accordingly.
Did the authors sample the feed to determine the microbiota present?
Thank you for your question. Unfortunately, these feed samples were not analyzed for 16S rRNA gene sequencing. However, we have been conducting other feeding trials in the same line of research, and using the same feed additives. The feed itself results in relatively low numbers of reads and the great majority (>60%) are identified as chloroplasts and mitochondria. Diets supplemented with the same probiotic resulted in~30% relative abundance of Pediococcus.
Did the authors detect the probiotc in the intestinal samples?
This study only analyzed samples from the digesta.
Minor:
427-430: This is a difficult conclusion to make without knowing the composition of the microbiota on the feed before feeding as this will influence the transient microbiome.
Thank you for your comment. As mentioned in the previous comment, our recent experience sequencing the 16S rRNA gene from the feed did not yield a high number of sequences, and most of them were identified as chloroplasts or mitochondria. For a study that we are currently conducting with the same feed additives, Pediococcus spp. was present at over 30% relative abundance in the feed supplemented with the commercial probiotic.
441-448: Without knowing the species it is difficult to claim that reduction in the relative abundance of Streptococcus is beneficial to the host. There are several species of Streptococcus that are part of the normal flora of the gut.
Thank you for your observation. We understand that our methods had limitations and we cannot accurately claim that the bacterial species were in fact pathogenic. However, when we access the identified bacteria on species level from the MiSeq data, we observed that for the control group the Streptococcus bacteria present in their transient digesta were Streptococcus agalactiae, S. iniae, and S. parauberis. The combination of these potential pathogenic bacteria summed ~10%, when compared to the probiotic (~3%), prebiotic (~1.2%), and Pre+Pro (~2.3%) groups. This information will be added to the discussion, and it will be highlighted that more refined, long sequencing methods are warranted to confirm this observation.
Reviewer 4 Report
MS presents the topic extensively. M&M are clerly explained and as well as results. Conclusion are in line with findings
Some minor comments:
Lines 90-91 2.1 paragraph
1) please specify the method of digestible energy estimation (by a bomb calorimeter or using a formula?)
2) report energy (MJ) for each diet also in the Table 1 where such information is not present
Line 133: and randomly too? Please specify
Lines 163-169: equations are poorly formatted
Author Response
MS presents the topic extensively. M&M are clerly explained and as well as results. Conclusion are in line with findings
Thank you for your kind comment. We hope to have addressed your questions and concerns, and we remain at your disposal for further clarification.
Some minor comments:
Lines 90-91 2.1 paragraph
Thank you for your suggestion. The format was performed by the MDPI template. We tried to adjust the formulas and we hope that they will be more acceptable.
1) please specify the method of digestible energy estimation (by a bomb calorimeter or using a formula?)
Thank you for your question. Modifications were made accordingly.
2) report energy (MJ) for each diet also in the Table 1 where such information is not present
Thank you for your comment. The calculated energy for the control diet should not be much different from the supplemented feeds; thus, the authors would like to keep table 1 as is.
Line 133: and randomly too? Please specify
Thank you for your question. Modifications were made accordingly.
Lines 163-169: equations are poorly formatted
Thank you for your observation. Modifications were made.
Round 2
Reviewer 2 Report
The revised version of the manuscript submitted by Yamamoto and colleagues has been reviewed. Unfortunately, although authors addressed satisfactorily some minor issues raised by this reviewer in the previous version of the manuscript, I still regret to consider that the present manuscript is not suitable for publication in Animals journal, at least on its present form. This is exclusively based on the major issues identified in the previous round of review process that authors didn’t address conveniently. Also, please, provide a manuscript version where each correction that has been made is highlighted in yellow. This would help to accelerate the speed of reviewing the next version.
Below are the major issues that still needs to be addressed:
1-Authors do not present adequately the results obtained from the present study. Authors did not include SD values for each evaluated parameter on each experimental group. Authors denied to present it arguing that other published papers did not present SD and were accepted (other journals than Animals). Independently of where, when and by who those articles were accepted, the present reviewer can provide thousands of articles were SD is reported. Thus, this is not an acceptable scientific argument, and the presentation of SD should be also included whether a two-way ANOVA and/or other statistical analysis is conducted. Indeed, this should be mandatory in order to show how variable is each evaluated parameter within the biological replicates, and how robust are the conclusions drawn from results. Furthermore, this is needed for providing “a concise and precise description of the experimental results”, as stated in the guidelines for authors that want to publish in Animals.
2-Regarding the immune response, authors did not evaluate the “response” of the immune system by a pathogenic challenge. To be clearer, the authors are evaluating the immunological status of the fish, but not the “response” against a pathogen or a molecule mimicking a pathogenic infection. Please, be accurate with the terms used. Although the present manuscript provides some information about the physiological status of the fish in this aspect, no conclusions can be made regarding the potential effect of the tested compounds to stimulate or inhibit the immune response against a pathogen. Please, it is a matter of accurateness on the use of concepts, avoid the use of immune response.
3-Although authors stated that “Additional information was added as requested”, present reviewer cannot identify it as included/modified text was not highlighted in the new version. Also, please, provide a brief description on how authors correct the descriptions in Mat and Met in your answer.
4-Resuls by DGGE and DNA-Seq are not consistent. In fact, results are contradictory. Authors describe the basis of each analytical method and that this might explain the differences in sensitivity between DGGE and MiSeq. The reviewer is aware of the pros and cons of both analyses, and thus what we could expect is that the results from both techniques have slight differences, but not contradictory results. Contradictory results are not “differences in sensitivity”. Furthermore, please, include the representation of all the replicates in both analyses.
5-Again, I kindly ask to authors to include the values from each biological replicate from each group in all the figures.
6-Regarding the conclusion regarding how Illumina platform MiSeq is hampering the ability to identify particular species seems to be inaccurate, the ability of species identification by NGS is not hampered by the length of sequenced amplicons by MiSeq, it was hampered by the pairs of primers used for amplifying the V4 region of the 16S rRNA gene. This region is no table to distinguishing all the genus/species, it has some limitations. The answer provided by authors confirms that they are not very familiar with this experimental approach and/or analytics.
Minor details:
L89-99: Again, which was the level of supplementation for Gro-biotic®-A? In table 1 is indicated 20 g/kg. But which is the rationale behind?
L156-157: Authors answer that the number of sampled fishes for the initial nutrient composition was not registered, since they used the biomass as the main parameter. Nevertheless, they didn’t explain the rationale behind the decision of doing this as whole-body composition instead of fillet composition. Again, the main point is the nutritional composition of what consumers eat. In addition, including several tissues, there is a chance that potential differences might be masked. Please, provide a convenient explanation for sampling whole-body composition.
L141-142: Another issue that has not been answered, authors evaluated if among experimental groups showed differences in feed intake?
L164: Since authors evaluated HIS and intraperitoneal fat, why no viscerosomatic index was evaluated?
L177-178: Regarding the little amount of fish included in each tank (12) with respect to the total fish sampled (9), and the potential effect of the decreased fish density at the end over the microbiome. Authors suggested that, if happening, “the impact on the microbiome be equally distributed among all dietary treatments”. Although this might be true, this might affect differentially the microbiome, being another factor altered that might compute for the overall result. Some experimental groups might cope with this effect equally or not, as they might cope equally or not with other stressors (e.g. the presence of a pathogen, etc.). Thus, the potential equal impact is highly speculative.
L182: Again, please, avoid self-citations. Yamamoto is not a technical paper where an analytical procedure is particularly described. Authors can cite the articles describing the procedure conducted in reference number 12.
L195: please, which is the rationale of using V3 region in the DGGE and the V4 region for Illumina MiSeq?
L204: reviewer insist that in addition to the bioproject, authors should present a table compiling the sequencing output including: number of raw reads, trimmed reads, GC content, sequencing quality scores, etc.... for each sample. How many samples were individually sequenced per experimental group? This table is mandatory to showed that the validity and robustness of the results here presented are not compromised by the quality of the sequencing and/or by the analysis of a limited number of biological replicates.
L228-231: Again, why not conducting Tukey if no interaction was found?
Student T-test is more appropriate when testing for two different treatments.
Again, Table 1: Final weight gain of probiotic group was higher, but showing lower survival rate. Feed addition was corrected considering this lower fish survival? Please, answer yes or not, and if not, how this might affect fish growth results.
Please, I kindly ask again, in Figure 4 and 5, to represent the 4 replicates from each group to clearly shown that all replicates are responding equally. A heatmap showing how the replicates from different experimental groups are clustering is required. This is not a matter of authors preference, this is a matter of showing reproducibility of the results in the different biological replicates conducted within each experimental group.
L370-372: again, if not providing any article supporting this statement, e.g. any article that has analyzed the activity of antinutritional factors or the activity of protease (e.g. trypsin), this is a mere speculation. Please, avoid it.
Author Response
N/A